# Role of Arginine Methylation in Alternative Polyadenylation of VEGFR-1 (Flt-1) pre-mRNA

**DOI:** 10.3390/ijms21186460

**Published:** 2020-09-04

**Authors:** Takayuki Ikeda, Hidehito Saito-Takatsuji, Yasuo Yoshitomi, Hideto Yonekura

**Affiliations:** Department of Biochemistry, Kanazawa Medical University School of Medicine, 1-1 Daigaku, Uchinada, Kahoku-gun, Ishikawa 920-0293, Japan; saitoh@kanazawa-med.ac.jp (H.S.-T.); yositomi@kanazawa-med.ac.jp (Y.Y.); yonekura@kanazawa-med.ac.jp (H.Y.)

**Keywords:** VEGFR-1, sVEGFR-1, alternative polyadenylation, poly(A) signal, arginine methylation, PRMTs, VEGF, angiogenesis

## Abstract

Mature mRNA is generated by the 3ʹ end cleavage and polyadenylation of its precursor pre-mRNA. Eukaryotic genes frequently have multiple polyadenylation sites, resulting in mRNA isoforms with different 3ʹ-UTR lengths that often encode different C-terminal amino acid sequences. It is well-known that this form of post-transcriptional modification, termed alternative polyadenylation, can affect mRNA stability, localization, translation, and nuclear export. We focus on the alternative polyadenylation of pre-mRNA for vascular endothelial growth factor receptor-1 (VEGFR-1), the receptor for VEGF. VEGFR-1 is a transmembrane protein with a tyrosine kinase in the intracellular region. Secreted forms of VEGFR-1 (sVEGFR-1) are also produced from the same gene by alternative polyadenylation, and sVEGFR-1 has a function opposite to that of VEGFR-1 because it acts as a decoy receptor for VEGF. However, the mechanism that regulates the production of sVEGFR-1 by alternative polyadenylation remains poorly understood. In this review, we introduce and discuss the mechanism of alternative polyadenylation of VEGFR-1 mediated by protein arginine methylation.

## 1. Introduction

Mature mRNA is generated by endonucleolytic cleavage of the 3ʹ end of pre-mRNA, followed by the synthesis of a polyadenosine tail. It is well understood that cleavage and polyadenylation require consensus sequences on pre-mRNA and more than 20 core proteins [1,2]. The polyadenylation machinery is composed of four subcomplexes: (1) cleavage and polyadenylation specificity factor (CPSF), which contains CPSF160 (CPSF1), CPSF100 (CPSF2), CPSF73 (CPSF3), CPSF30 (CPSF4), WDR33, and factor interacting with poly(A) polymerase (FIP1); (2) cleavage stimulation factor (CSTF), which contains CSTF55 (CSTF1), CSTF64 (CSTF2), and CSTF77 (CSTF3); (3) cleavage factor I (CFIm), which contains CFIm25 (CPSF5), CFIm68 (CPSF6), and/or CFIm59 (CPSF7); and (4) cleavage factor II (CFIIm), which contains PCF11 and CLP1. The CPSF complex interacts with the polyadenylation (poly(A)) signal (typically AAUAAA) through CPSF30 and WDR33. CFIm25 and CSTF64 bind the UGUA element and the U- and GU-rich elements that are upstream and downstream of the poly(A) signal, respectively. CPSF73 is an endonuclease that cleaves approximately 20 nt downstream of the poly(A) site, while poly(A) polymerase synthesizes the poly(A) tail.

Almost all eukaryotic genes have multiple poly(A) sites, which enable the formation of distinct mRNA isoforms by the differential usage of these sites; this is termed as alternative polyadenylation (APA) (Figure 1) [3,4]. Most APA sites are located in the same terminal exon (3ʹ-UTR), resulting in mRNA isoforms with differing 3ʹ-UTR lengths. As 3ʹ-UTRs contain regulatory sequences that are involved in various aspects of mRNA metabolism, APA can affect mRNA stability, localization, and translation, and lead to differences in the levels of proteins encoded by these genes [1]. Furthermore, many APA sites are located in introns. This type of APA leads to the expression of alternative terminal exons, resulting in different C-terminal coding sequences and 3ʹ-UTRs in mRNAs. Alternative terminal exons can be divided into two subtypes. Composite terminal exons are generated by using the adjacent intron as the last exon via inhibition of the 5ʹ splice site, whereas alternative splicing and polyadenylation in internal introns give rise to cassette terminal exons. A recent study reported that approximately 10% of the experimentally identified poly(A) sites are located in introns [5]. A well-known example of the type of composite terminal exons generated by APA is the immunoglobulin M (IgM) heavy chain. The poly(A) site of IgM mRNA can be switched from a distal poly(A) signal in the 3ʹ-most terminal exon to a proximal poly(A) signal in a composite terminal exon during B cell activation [6,7]. Furthermore, it was reported that secreted isoforms of receptor tyrosine kinases (RTKs) are produced by APA [8]. The EST (expressed sequence tag) database was queried by intronic sequences in the 5ʹ region around or upstream of the exon encoding the transmembrane domain, following the identification of 31 mRNA variants from 19 of 24 RTK genes. Interestingly, soluble VEGFR-2 (sVEGFR-2) is encoded by a VEGFR-2 mRNA isoform that is produced by cleavage and polyadenylation in intron 13 of the VEGFR-2 pre-mRNA, identically to VEGFR-1 [9,10]. In this review, we will focus on the mechanism of sVEGFR-1 production.

## 2. Alternative Polyadenylation for sVEGFR-1 Production

### 2.1. VEGFR-1 Splice Variants

Angiogenesis is the process of new blood vessel formation from pre-existing capillaries and plays a significant role in the progression of diseases such as cancer, diabetic retinopathy, and rheumatoid arthritis [11,12]. Angiogenesis is strongly regulated by the balance between pro- and anti-angiogenic molecules [13]. VEGF is a crucial regulator of angiogenesis under physiological and pathological conditions [11,14]. VEGF, also known as VEGF-A, stimulates endothelial cell migration and proliferation and induces angiogenesis by binding to specific receptors: VEGFR-1 (Flt-1; fms-like tyrosine kinase-1) and VEGFR-2 (KDR; kinase insert domain-containing receptor/Flk-1; fetal liver kinase-1) [14]. Although VEGFR-1 binds to VEGF with a higher affinity than that of VEGFR-2, the primary receptor for VEGF [15], the roles of VEGFR-1 as a transmembrane receptor in the context of endothelial cell function are poorly understood. Moreover, the soluble form of VEGFR-1 (sVEGFR-1 or sFlt-1), which contains the extracellular VEGF-binding domain, is secreted into the extracellular space [16]. It is well-known that sVEGFR-1 functions as a decoy receptor by trapping VEGF, thus serving as a potent endogenous anti-angiogenic factor [17,18]. The short form of VEGFR-1 is generated by APA at the mRNA level. 

The VEGFR-1 gene consists of 30 exons and encodes seven extracellular Ig-like domains, a single transmembrane domain, and an intracellular tyrosine kinase domain. Six VEGFR-1 mRNA variants have been reported to date (Figure 2A) [19,20,21]. The longest form encodes membrane-bound VEGFR-1 (mVEGFR-1), which has kinase activity and can activate specific signaling cascades in response to ligand binding [14]. All other forms encode truncated proteins that lack the transmembrane and intracellular kinase domains, resulting in the secreted (soluble) form of VEGFR-1. sVEGFR-1_i13 was initially identified as a 100-kDa protein expressed in vascular endothelial cells by Kendall et al. [16]. The sVEGFR-1_i13 protein contains the first six Ig-like domains and 31 unique C-terminal amino acids. The 687-amino-acid sVEGFR-1_i13 protein is encoded by the sVEGFR-1_i13S and sVEGFR-1_i13L mRNAs, both of which are composed of the first 13 exons and part of the intron 13 sequence but with either a 17 or 4146 nt 3ʹ-UTR, respectively (Figure 2A) [22]. sVEGFR-1_i13S is the predominant isoform in human microvascular endothelial cells (HMVEC) (Figure 2B) and is a major isoform in various tissues [23,24,25].

sVEGFR-1_i14 mRNA contains the first 14 exons and intron 14 and encodes the 736 amino acid sVEGFR-1_i14 protein. sVEGFR-1_e15a and sVEGFR-1_e15b result from alternative splicing by using two different cryptic splice acceptor sites, yielding new exons (15a and 15b, respectively) and termination at alternative polyadenylation sites in intron 14 [23,26]. sVEGFR-1_e15a and sVEGFR-1_e15b proteins contain specific C-terminal amino acids, 28 and 13 amino acids, respectively, in addition to the 705 amino acids of the first 14 exons. sVEGFR-1_e15a is the most abundant isoform in the placenta [25,27,28]. 

### 2.2. sVEGFR-1 Production by Alternative Polyadenylation (APA)

The sVEGFR-1_i13 mRNA variant was first identified in a human vascular endothelial cell cDNA library [16]. The sVEGFR-1_i13 mRNA contains intron 13 sequences, suggesting that it was generated by read-through splice skipping at the splice site of intron 13. Thereafter, multiple cleavage sites for sVEGFR-1_i13 production were identified by a 3ʹRACE assay of a placental cDNA library [22]. One sVEGFR-1_i13 cleavage site was located 21–29 nt downstream of the most proximal poly(A) signal in intron 13, resulting in a short (~20 nt) 3ʹ-UTR (sVEGFR-1_i13S in Figure 2A). sVEGFR-1_i13L uses the distal poly(A) signals in intron 13, and consequently the 3ʹ-UTR is relatively long (~4.1 kb) (Figure 2A). We also identified the cleavage site of sVEGFR-1_i13 in HMVEC using a cDNA library [24] and a 3ʹRACE protocol (Figure 2B). In HMVEC, the major cleavage site was the most proximal site, consistent with a previous report (Figure 2B). These results indicate that VEGFR-1 pre-mRNA is processed at nucleotide position 112 of intron 13 (Figure 2C) and that sVEGFR-1_i13S is expressed as the predominant form in HMVEC [24]. Therefore, it was demonstrated that sVEGFR-1_i13 arises from the same gene that encodes VEGFR-1 (mVEGFR-1) by APA within intron 13. Henceforth, sVEGFR-1_i13S mRNA and protein will be denoted by sVEGFR-1.

### 2.3. Regulation of VEGFR-1 APA

The extracellular and intracellular stimuli that regulate VEGFR-1 APA are largely unknown, but some factors have been reported. Hypoxia regulates sVEGFR-1 expression, although this effect is somewhat controversial because it differs among different tissues and cells. The effect of the oxygen concentration on the APA of VEGFR-1 in HMVEC has been reported [24]. sVEGFR-1 expression is downregulated by hypoxic conditions (1%–5% O_2_), whereas mVEGFR-1 expression remains unchanged, suggesting that the oxygen concentration can specifically modulate sVEGFR-1 expression. Furthermore, the sVEGFR-1/mVEGFR-1 mRNA ratio is not impacted by treatment with DMOG, an inhibitor of prolyl hydroxylase (PHD) that stabilizes hypoxia-inducible factor-1α (HIF-1α) under hypoxic conditions. These results indicated that hypoxia specifically regulates the levels of sVEGFR-1 mRNA, and thus, the APA of VEGFR-1 is independent of HIF-1α in HMVEC. In contrast, Boeckel et al. argued that hypoxia upregulates sVEGFR-1 expression [29]. Severe hypoxic conditions (0.1% O_2_) upregulate sVEGFR-1 expression mediated by jumonji domain-containing 6 (Jmjd6) protein and the U2 small nuclear ribonucleoprotein auxiliary factor 65 kDa subunit (U2AF65) in human umbilical vein endothelial cells (HUVEC). Chronic hypoxia (2%, 72 h) also results in significant increases in sVEGFR-1 mRNA and protein expression in cytotrophoblast cells [30]. This discrepancy might be due to the difference in cell specificity between the microvasculature (HMVEC) and macrovasculature (HUVEC). The mechanisms behind APA regulation remain unclear, though hypoxia can change VEGFR-1 expression at the mRNA level. Moreover, VEGF was also reported to regulate VEGFR-1 APA. In cancer cells, VEGF165, a splice variant of VEGF-A, upregulates sVEGFR-1 in cooperation with the transcription factor SRY-box transcription factor 2 (SOX2) and the splicing factor serine/arginine-rich splicing factor 2 (SRSF2) [31]; however, VEGF165 has no effect on sVEGFR-1 production in HMVEC [24]. This suggests the potential existence of tissue and cell type-specific mechanisms for VEGFR-1 APA. Other factors that increase sVEGFR-1 release have been reported by other research groups [32,33,34,35], but whether they regulate VEGFR-1 APA remains unclear.

### 2.4. Regulatory Sequences for APA of VEGFR-1

The cleavage and polyadenylation sites of pre-mRNA are primarily defined by the hexameric consensus motif AAUAAA, which is the canonical poly(A) signal [2]. Similar sequences (e.g., AUUAAA, AGUAAA, UAUAAA) also function as poly(A) signals [36]. Intron 13 of the VEGFR-1 gene contains five canonical (AAUAAA) and six putative (AUUAAA) poly(A) signals. Among them, the most proximal putative two poly(A) signals (AUUAAA) are located upstream of the major proximal cleavage site described above (Figure 2C). We investigated which signal is involved in cleavage and polyadenylation at the proximal site by using a VEGFR-1 minigene in HMVEC [24]. The minigene contains exons 12–14 of the VEGFR-1 gene, excluding the internal segments of introns 12 and 13. In this experimental APA system, the soluble form of the RNA containing exon 12, exon 13, and intron 13 is expressed when the APA site within intron 13 is used. If the introns are spliced out, the membrane form of the RNA, which comprises exons 12, 13, and 14, is generated. We confirmed that the soluble form of RNA from the minigene was cleaved and polyadenylated at nucleotide position 112 of intron 13, which corresponds to the endogenous cleavage site, suggesting that the minigene mimics the endogenous APA of VEGFR-1 in HMVEC. When the upstream AUUAAA sequence was mutated to AUCCCA, the ratio of the soluble form to the membrane form decreased to approximately 10%, and the soluble form of RNA was completely abolished by mutating both AUUAAA sequences. These results indicated that the upstream AUUAAA sequence functions as a major poly(A) signal for sVEGFR-1 mRNA production and as a complete poly(A) signal that synergizes with the downstream AUUAAA sequence [24]. 

In addition to the proximal poly(A) signal, there are three canonical poly(A) signals positioned upstream of distal cleavage sites at the 3ʹ region of intron 13 (~4 kb downstream from the proximal cleavage site). Another group identified a primary poly(A) signal at the distal cleavage sites, using a poly(A) signal reporter vector in cytotrophoblasts from placenta [22]. Their experiments showed that one canonical poly(A) signal produced the longest mRNA, while another noncanonical poly(A) signal (UAUAAA) is used for an mRNA that is slightly shorter than the longest form. It is known that the canonical poly(A) signal is enriched at the distal portion of 3ʹ-UTR, compared to the juxtaposition of the stop codon [37]. The VEGFR-1 gene intron 13 contains canonical (AAUAAA) sites in the distal portion of the intron according to the typical tendency; however, it remains unclear how the poly(A) site usage is regulated.

Cleavage and polyadenylation are executed by the 3ʹ end processing complex. The proximal region of intron 13 of VEGFR-1 contains RNA sequences recognized by the 3ʹ end processing complex, such as the U-rich and G/U-rich regions around the cleavage site (Figure 2C) [24]. Therefore, it is expected that RNA-binding proteins that recognize the specific motif might regulate the splicing and/or cleavage and polyadenylation of VEGFR-1 intron 13; thus, the putative recognition motifs of RNA-binding proteins could be potential regulatory elements. Well-characterized RNA-binding proteins include serine/arginine (SR) proteins and AU-rich element (ARE)-binding proteins. Notably, we reported that one ARE located downstream of the sVEGFR-1 cleavage site regulates sVEGFR-1 mRNA production [24]. This sequence is part of the putative ARE (AUUUA) located 24 nt downstream of the major cleavage site. The replacement of UUU by AAA at positions 137–139 nt in intron 13 led to a significant decrease in the soluble form of RNA in a minigene assay, suggesting that ARE-binding proteins may function as regulators of VEGFR-1 APA.

### 2.5. Regulatory Factors for APA of sVEGFR-1

The core polyadenylation machinery components are important for APA regulation. Higher CSTF64 protein levels during B cell activation lead to the usage of the upstream intronic poly(A) site in the IgM heavy chain transcript [38]. The upregulation of CSTF64 promotes the usage of proximal poly(A) sites, thereby preventing usage at the distal poly(A) sites [39,40,41]. A hypoxic condition that upregulated sVEGFR-1 expression in HMVEC did not induce an increase in CSTF64. Additionally, CSTF64 overexpression in HMVEC did not accelerate VEGFR-1 APA (Ikeda et al., unpublished data) [24], suggesting that CSTF64 protein levels are not involved in the proximal cleavage and polyadenylation of sVEGFR-1. CFIm25 (CPSF5), which recognizes the UGUA motif upstream of the poly(A) signal, is also involved in the selection of the poly(A) site. CFIm25 knockdown was reported to lead to the use of upstream poly(A) signals and the shortening of the 3ʹ-UTR [37,42,43,44], and the overexpression of CFIm25 inhibits tumorigenicity and tumor growth [44]. However, CFIm25 expression was not implicated in the regulation of VEGFR-1 APA by hypoxia in HMVEC (Ikeda et al., unpublished data) [24]. 

RNA-binding proteins are known to be APA regulators. Splicing factors such as U1 snRNP, polypyrimidine tract-binding protein 1 (PTBP1), embryonic lethal abnormal vision-like protein 1 (ELAVL1, also known as HuR), heterogeneous nuclear ribonucleoprotein C (hnRNP C), and poly(C)-binding protein 1 (PCBP1) were shown to be involved in APA [2]. U2AF65 is necessary for splicing because it binds to the polypyrimidine tract of the 3ʹ portion of an intron. Furthermore, U2AF65 regulates cleavage and polyadenylation by interacting with the CFI complex [45]. In a previous study, the binding of U2AF65 to sVEGFR-1 mRNA could mediate VEGFR-1 APA by interacting with JMJD6 under hypoxia [29,46]. In contrast, another group demonstrated that JMJD6 and U2AF65 are not necessary or have little effect on sVEGFR-1 production [47]. The roles of these proteins in VEGFR-1 APA therefore remain controversial. 

The AUUUA sequence is the candidate regulatory motif used to generate sVEGFR-1 by APA, as described above (Figure 2C). Several proteins that recognize AU-rich sequences in mRNAs are known. hnRNP D (also known as AUF1) is an ARE-binding protein implicated in the regulation of mRNA splicing and stability [48]. We found that hnRNP D binds to VEGFR-1 pre-mRNA, and hnRNP D overexpression dramatically decreased sVEGFR-1 mRNA expression in HMVEC; furthermore, hnRNP D knockdown leads to a significant increase in sVEGFR-1 expression, indicating that VEGFR-1 APA is regulated by hnRNP D [49]. Other ARE-binding proteins that recognize AUUUA (HuR, hnRNP A2/B1, hnRNP A0, TIA1, TIAL1) did not affect the APA of VEGFR-1 (Ikeda et al., unpublished data).

## 3. Arginine Methylation on the APA of VEGFR-1

### 3.1. Protein Arginine Methylation 

Arginine methylation is a common post-translational modification that affects transcription, RNA splicing, mRNA translation, DNA repair, and signal transduction [50,51]. This process is carried out by a nine-member family of enzymes termed protein arginine methyltransferases (PRMTs). PRMTs catalyze the transfer of a methyl group from the methyl donor *S*-adenosylmethionine (SAM) to a guanidino nitrogen atom. Three types of methylarginines have been identified in eukaryotes, namely ω-*N*^G^-monomethylarginine (MMA), ω-*N*^G^,*N*^G^-asymmetric dimethylarginine (aDMA), and ω-*N*^G^,*N*^G^- symmetric dimethylarginine (sDMA). PRMTs are classified into three groups according to their catalytic activity: type I enzymes (PRMT1–4, 6, and 8), which catalyze MMA and aDMA formation; type II enzymes (PRMT5 and 9), which are responsible for MMA and sDMA production; and a type III enzyme (PRMT7), which catalyzes only the formation of MMA. Arginine residues in glycine–arginine-rich motifs (RGG/RG motifs) are the main targets of arginine methylation by PRMTs. In terms of RNA metabolism, PRMT4 is involved in regulating pre-mRNA alternative splicing by methylating the splicing factors [52]. PRMT5 is a master regulator of splicing, as it regulates the methylation of spliceosomal Sm proteins [53]. PRMT9 can methylate splicing factors (SAP145 and SF3B2) to regulate alternative splicing [54,55]. 

Compared to PRMTs, arginine demethylase remains unclear [56]. JMJD6 targets methylated arginine residues; however, the role of JMJD6 as an arginine demethylase remains controversial because of the absence of evidence regarding direct demethylation activity. 

### 3.2. Involvement of Protein Arginine Methylation in the APA of VEGFR-1

We first reported the relationship between protein arginine methylation and VEGFR-1 APA using the global methyltransferase inhibitors, 5ʹ-methylthioadenosine (MTA) and adenosine dialdehyde (AdOx) [49]. The suppression of methylation in HMVEC by inhibitors induced the upregulation of sVEGFR-1 mRNA and downregulation of mVEGFR-1 mRNA, with a concomitant increase in the sVEGFR-1/mVEGFR-1 ratio (Figure 3A). We also demonstrated that PRMT1 overexpression dramatically decreased sVEGFR-1 mRNA, and that PRMT1 knockdown slightly increased the sVGEFR-1/mVEGFR-1 ratio (Figure 3B). Although it remains unknown whether PRMT1 is a major methyltransferase in HMVEC, these results suggest that protein arginine methylation may be involved in the APA of VEGFR-1 in HMVEC. Recently, Ishimaru et al. reported that vascular endothelial cell-specific PRMT1-deficient mice resulted in embryonic lethality due to poor development of the vascular network [57]. Considering that sVEGFR-1 overexpression resulted in vascular defects [58], arginine methylation (likely by PRMT1) may be involved in the regulation of vascular formation via the APA of VEGFR-1. In addition to PRMT1, arginine methyltransferases (PRMT4, PRMT5, and PRMT9) are involved in the regulation of splicing [52,53,54,59], but their functions in VEGFR-1 APA remains unknown.

hnRNP D was shown to be involved in VEGFR-1 APA, as described above [49]. hnRNP D has four isoforms (p45, p42, p40, and p37), and all isoforms contain three putative RGG/RG motifs in the C-terminal region. One of these motifs (R277 of p37 isoform) was reported to serve as a di-methylation site, as determined using a heavy SILAC method [60]. Another report showed that two motifs (R253 and R263 of p37 isoform) are monomethylated and could be further dimethylated by LPS treatment in macrophages [61]. Moreover, hnRNP D can be methylated by PRMT1 [52,62]. Therefore, we examined and demonstrated using the minigene assay that an RGG/RG motif mutant in hnRNP D (R277A) resulted in a slight but significant increase in the ratio of soluble form to membrane form of the RNA, whereas hnRNP D overexpression dramatically decreased the level of the soluble form [49]. Furthermore, the overexpression of JMJD6 slightly increased the ratio of soluble form to membrane form of the RNA, as determined using a minigene assay [49]. Interestingly, a synthesized peptide that includes the three RGG domains in the C-terminus of hnRNP D induced sVEGFR-1 expression [63], suggesting that the peptide may sequester arginine methyltransferase, which methylates endogenous hnRNP D. These results and reports suggest that unmethylated hnRNP D may act as an inducer of VEGFR-1 APA. 

Taken together, we propose a regulatory model of VEGFR-1 APA by hnRNP D and arginine methylation (Figure 3C). We hypothesize that hnRNP D may mask regulatory sequences (ARE) in the intron 13 sequence of VEGFR-1 pre-mRNA and competes with the polyadenylation machinery that interacts with the poly(A) signal and the surrounding requisite elements. The RNA-binding activity of hnRNP D is regulated by arginine methylation, which could be catalyzed by PRMT1 or other methyltransferases. PRMT1 activity may be stimulated by a low oxygen concentration, although this remains to be experimentally confirmed. Although more studies are needed to clarify the importance of arginine methylation in VEGFR-1 APA, we believe that our work will provide new clues for further clarifying the mechanisms involved in regulating pre-mRNA APA.

## Figures and Tables

**Figure 1 ijms-21-06460-f001:**
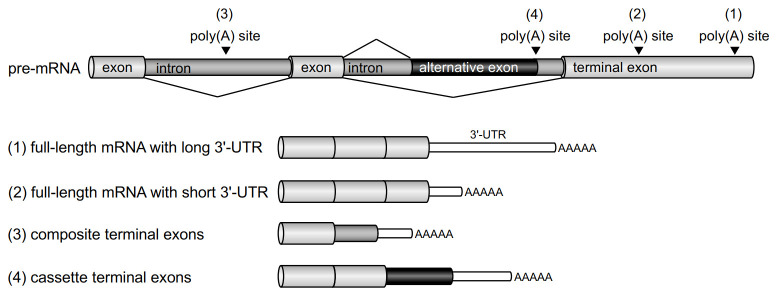
Different mRNA isoforms generated by alternative polyadenylation. Splicing is indicated by the lines in pre-mRNA diagrams. The 3ʹ-UTRs of mRNA isoforms are depicted by white boxes. Additional or alternative coding sequences of mRNA isoforms are shown by black boxes. (1, 2) Poly(A) signals in the terminal exon are used to generate full-length mRNAs with differing 3ʹ-UTR lengths. The full-length mRNA isoform with a longer 3ʹ-UTR (1) contains more regulatory elements than the isoform with a shorter 3ʹ-UTR (2). (3) The inhibition of splicing of an upstream intron results in the inclusion of the adjacent intron and the use of a poly(A) signal within introns. (4) The alternative splicing and usage of alternative poly(A) signals in the alternative exon generates a transcript containing different terminal exon and 3ʹ-UTR sequences.

**Figure 2 ijms-21-06460-f002:**
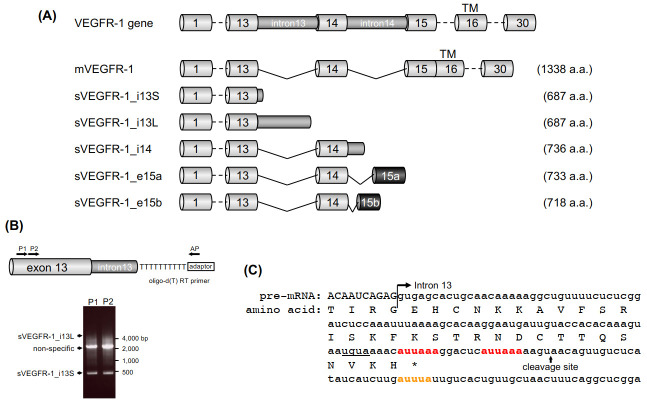
(**A**) Schematic representation of VEGFR-1 mRNA isoforms. Exons and introns are shown in light and dark gray, respectively. Black boxes (e15a and e15b) are new exons resulting from alternative splicing. Lines indicate the spliced introns. TM: transmembrane domain. (**B**) 3ʹRACE of VEGFR-1 mRNA variants in HMVEC. Poly(A) RNAs from HMVEC were reverse-transcribed using oligo-d(T) RT primer with adaptor sequence. PCR was performed using two primers (P1 and P2) targeting exon 13, which were paired with an adaptor primer (indicated by AP). The lengths of expected PCR products were approximately 420 bp for sVGEFR-1_i13S and 4.6 kbp for sVGEFR-1_i13L. (**C**) Nucleotide and amino acid sequences of sVGEFR-1 pre-mRNA around the exon 13–intron 13 junction of the gene encoding VEGFR-1. Exon and intron sequences are indicated by uppercase and lowercase symbols, respectively. The putative poly(A) signals are indicated in red. The putative ARE (AU-rich element), which regulates VEGFR-1 APA, is in orange. The underline represents the expected CFI complex-binding sequence, UGUA.

**Figure 3 ijms-21-06460-f003:**
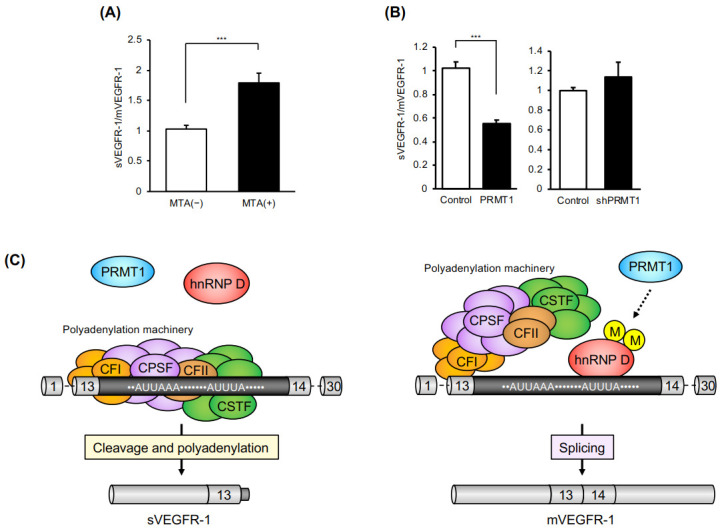
(**A**) Effect of methyltransferase inhibitor (MTA) on VEGFR-1 APA. HMVECs were treated with MTA (100 µM) for 24 h. Total RNAs were collected and purified with TRI Reagent and an RNeasy Mini Kit. The mRNA levels of sVEGFR-1 and mVEGFR-1 were determined by qRT-PCR using the following primers: 5′-TTGGGACTGTGGGAAGAAAC-3′ and 5′-TTGGAGATCCGAGAGAAAACA-3′ for sVEGFR-1, and 5′-CTTCACCTGGACTGACAGCA-3′ and 5′-TAGATGGGTGGGGTGGAGTA-3′ for mVEGFR-1. Data are expressed as means ± standard deviations (S.D.). *** *p* < 0.001. −: control, +: MTA treatment. (**B**) PRMT1 overexpression and knockdown affected the ratio of sVEGFR-1 to mVEGFR-1 in microvascular endothelial cells. HMVECs were infected by lentivirus particles, and total RNAs were purified after 48 h. qRT-PCR was performed using the primers described in (A). Data are expressed as means ± S.D. *** *p* < 0.001. (**C**) A regulatory model of alternative polyadenylation of VEGFR-1. Recognition of the poly(A) signals in intron 13 of the VEGFR-1 pre-mRNA via the polyadenylation machinery induces cleavage and polyadenylation in intron 13 following the production of sVEGFR-1 mRNA (left). The methylated hnRNP D (likely through PRMT1) binds to ARE (AUUUA), which may be recognized by the CSTF complex, and then sequesters the polyadenylation machinery. Consequently, intron 13 is spliced out by the splicing machinery, resulting in the production of mVEGFR-1 mRNA (right).

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
