# Peer review of "Role of Arginine Methylation in Alternative Polyadenylation of VEGFR-1 (Flt-1) pre-mRNA"

_ijms, 2020, doi:10.3390/ijms21186460_

Round 1

Reviewer 1 Report

This review summarized recent findings for the regulation of sVEGFR-1 by alternative polyadenylation via arginine methylation of hnPNP D by PRMT1. My major concern is as it is a review paper, instead of an original research article. It is not appropriate to present their data as shown at Figure 2 and 3. No information for the sequence of primers were used and details of the experiments (such as cell types and details of treatments) for reviewers to evaluate the findings and the readers to repeat the experiment. Suggest the authors resubmit this work as an original research article with adding the section "Methods and Materials".

Author Response

Responses to the comments of Reviewers:
Our point-by-point responses to the reviewers’ comments are provided below. The newly added and/or revised text is indicated in red in the revised manuscript. According to the comments made by Reviewer 1, we have revised the manuscript to be more suitable as a review. As per the comments made by Reviewer 2, we have added a new Figure 1 that summarizes the APA. In addition, we carefully reviewed the entire manuscript and improved the writing.

[Comment]
This review summarized recent findings for the regulation of sVEGFR-1 by alternative polyadenylation via arginine methylation of hnPNP D by PRMT1. My major concern is as it is a review paper, instead of an original research article. It is not appropriate to present their data as shown at Figure 2 and 3. No information for the sequence of primers were used and details of the experiments (such as cell types and details of treatments) for reviewers to evaluate the findings and the readers to repeat the experiment. Suggest the authors resubmit this work as an original research article with adding the section "Methods and Materials".
[Response]
Thank you very much for this constructive comment. We agree and have corrected the manuscript to be more suitable as a review. We have eliminated the data figures (original Figure 2B and 2C, Figure 3C) and described these findings only in the text. Additionally, the original Figure 2A (pre-mRNA sequence) is included as Figure 2C in the revised Figure 2. We would like to present Figure 3A and 3B to improve the reader’s understanding because these data (ref. 49) remain the only evidence suggesting the involvement of arginine methylation in the VEGFR-1 APA. We believe that this is the key point behind our claim of a new model in this review. We hope that this correction will meet with your approval, as the retention of these data would help the readers to better understand our hypothesis.
Per your comments, we have added information about the primer sequences and experimental details in Figure 3A and 3B to the Figure Legend (lines 290–299) as follows: “(A) Effect of methyltransferase inhibitor (MTA) on VEGFR-1 APA [49]. HMVECs were treated with MTA (100 µM) for 24 h. Total RNAs were collected and purified with TRI Reagent and an RNeasy Mini Kit. The mRNA levels of sVEGFR-1 and mVEGFR-1 were determined by qRT-PCR using the following primers: 5′-TTGGGACTGTGGGAAGAAAC-3′ and 5′-TTGGAGATCCGAGAGAAAACA-3′ for sVEGFR-1, and 5′-CTTCACCTGGACTGACAGCA-3′ and 5′-TAGATGGGTGGGGTGGAGTA-3′ for mVEGFR-1. Data are expressed as means ± standard deviations (S.D.). ***p < 0.001. (B) PRMT1 overexpression and knockdown affected the ratio of sVEGFR-1 to mVEGFR-1 in microvascular endothelial cells. HMVECs were infected by lentivirus particles, and total RNAs were purified after 48 h. qRT-PCR was performed using the primers described in (A). Data are expressed as means ± S.D. ***p < 0.001. ”.

Reviewer 2 Report

The authors described the molecular mechanisms of VEGFR-1 APA. In particular, methylation of hnRNP D is important for the APA. The manuscript is informative and well written. However, I feel that following points need to be addressed to enhance the importance of the manuscript.

  • In line 44, the authors described “…APA can affect mRNA stability, localization, translation, and nuclear export…”. Is the localization different from nuclear export? Or does this mean different nuclear bodies?
  • In line 45, I could not catch the meaning of the phrase “due to differences in protein levels”. What is the protein? 3’ UTR binding proteins?
  • On page 2, the authors described the molecular mechanism of APA. I think an additional, generalized figure regarding APA mechanism helps the readers to understand the mechanism.
  • In line 53, is APA caused by intron retention?
  • In line 87, does sVEGFR-1 mean the soluble protein which is translated from sVEGFR-1_i13S and L? Even though the authors wrote “sVEGFR-1_i13 (henceforth called sVEGFR-1)”, there are some “sVEGFR-1_i13 protein” after that. In addition, I found some sVEGFR-1 mRNA throughout the manuscript. I think this is quite confusing. The authors should consider changing way of description.

Minor points

  • In line 35, the major form of PAS is AAUAAA in my knowledge.
  • In line 37, CPSF73, instead of CPST73.
  • In line 74, VEGFR-2, instead of VEGR-2.
  • In line 145, is VEGF165 the only VEGF variant upregulates sVEGFR-1? How about the other VEGFs?

Author Response

Responses to the comments of Reviewers:
Our point-by-point responses to the reviewers’ comments are provided below. The newly added and/or revised text is indicated in red in the revised manuscript. According to the comments made by Reviewer 1, we have revised the manuscript to be more suitable as a review. As per the comments made by Reviewer 2, we have added a new Figure 1 that summarizes the APA. In addition, we carefully reviewed the entire manuscript and improved the writing.

[Comment]
In line 44, the authors described “...APA can affect mRNA stability, localization, translation, and nuclear export...”. Is the localization different from nuclear export? Or does this mean different nuclear bodies?
In line 45, I could not catch the meaning of the phrase “due to differences in protein levels”. What is the protein? 3’ UTR binding proteins?
[Response]
We have corrected this statement as, “APA can affect mRNA stability, localization, and translation, and lead to differences in the levels of proteins encoded by these genes” in the revised manuscript (line 45).

[Comment]
On page 2, the authors described the molecular mechanism of APA. I think an additional, generalized figure regarding APA mechanism helps the readers to understand the mechanism.
[Response]
Thank you for your suggestion. Accordingly, we have added a new Figure 1 that depicts the general APA mechanism to the revised manuscript. The original Figure 1 has been reorganized into Figure 2 in the revised manuscript.

[Comment]
In line 53, is APA caused by intron retention?
[Response]
Thank you for your careful review and comment. We have eliminated the sentence because we misread the paper cited in our original manuscript.

[Comment]
In line 87, does sVEGFR-1 mean the soluble protein which is translated from sVEGFR-1_i13S and L? Even though the authors wrote “sVEGFR-1_i13 (henceforth called sVEGFR-1)”, there are some “sVEGFR-1_i13 protein” after that. In addition, I found some sVEGFR-1 mRNA throughout the manuscript. I think this is quite confusing. The authors should consider changing way of description.
[Response]
Thank you for your advice. Accordingly, we have unified the description of sVEGFR-1 in Sections 2.1 and 2.2. Furthermore, we have added the sentence, “Henceforth, sVEGFR-1_i13S mRNA and protein will be denoted by sVEGFR-1,” to represent the use of sVEGFR-1 instead of sVEGFR-1_i13 at the end of Section 2.2.

[Comment]
In line 35, the major form of PAS is AAUAAA in my knowledge.
In line 37, CPSF73, instead of CPST73.
In line 74, VEGFR-2, instead of VEGR-2.
[Response]
We have corrected the mistakes in the revised manuscript.

[Comment]
In line 145, is VEGF165 the only VEGF variant upregulates sVEGFR-1? How about the other VEGFs?
[Response]
Faycal et al. reported that sVEGFR-1-i13 mRNA but not VEGFR-1 mRNA was increased upon treatment with only VEGF165 (ref. 31). The authors treated lung carcinoma cells for 24 hours with VEGF121, VEGF165, and VEGF189, and VEGF121 and VEGF189 did not increase the expression of sVEGFR-1-i13 mRNA. However, the results may differ between cells and tissues. In our report (ref. 24), VEGF165 had no effect on sVEGFR-1 production in HMVECs, as described in line 164 of the revised manuscript.

Round 2

Reviewer 1 Report

I have no further question.

Reviewer 2 Report

I am satisfied with the revised manuscript.